# Prognostic Impact of Lymphatic Invasion in Patients with High-Risk Prostate Cancer after Robot-Assisted Radical Prostatectomy and Extended Lymph Node Dissection: A Single-Institution Prospective Cohort Study

**DOI:** 10.3390/cancers14143466

**Published:** 2022-07-17

**Authors:** Shimpei Yamashita, Satoshi Muraoka, Takahito Wakamiya, Kazuro Kikkawa, Yasuo Kohjimoto, Isao Hara

**Affiliations:** Department of Urology, Wakayama Medical University, 811-1 Kimiidera, Wakayama 641-0012, Japan; keito608@wakayama-med.ac.jp (S.Y.); muraoka@wakayama-med.ac.jp (S.M.); wakataka@wakayama-med.ac.jp (T.W.); kzro@wakayama-med.ac.jp (K.K.); hara@wakayama-med.ac.jp (I.H.)

**Keywords:** prostate cancer, lymphatic invasion, biochemical recurrence, robot-assisted radical prostatectomy

## Abstract

**Simple Summary:**

The aim of our prospective cohort study was to assess the impact of lymphatic invasion on biochemical recurrence (BCR) in patients who underwent robot-assisted radical prostatectomy (RARP) and extended lymph node dissection (eLND) for high-risk prostate cancer (PC). Of 183 patients, lymphatic invasion and lymph node metastasis were observed in 47 (26%) and 17 patients (9%), respectively, whereas BCR was observed in 48 patients (26%). The BCR rate was significantly higher in patients with lymphatic invasion than in patients without lymphatic invasion. Moreover, according to multivariable analyses, lymphatic invasion was an independent significant predictor of BCR in the overall patient group and in patients without lymph node metastasis. Evaluation of lymphatic invasion could therefore be a useful predictor of BCR in patients who have undergone RARP and eLND for high-risk PC.

**Abstract:**

The prognostic impact of lymphatic invasion in patients with high-risk prostate cancer (PC) remains unclear. The aim of our single-institution prospective cohort study was to examine the impact of lymphatic invasion on biochemical recurrence (BCR) in patients with high-risk PC according to National Comprehensive Cancer Network (NCCN) criteria who underwent robot-assisted radical prostatectomy (RARP) and extended lymph node dissection (eLND). A total of 183 patients were included who underwent RARP and eLND for NCCN high-risk PC between June 2014 and August 2019. Lymphatic invasion in resected specimens was observed in 47 patients (26%), whereas lymph node metastasis was observed in 17 patients (9%). During follow-up, BCR was observed in 48 patients (26%). The BCR rate in patients with lymphatic invasion was significantly higher than that in patients without lymphatic invasion (*p* < 0.01). According to multivariable Cox proportional hazards regression analyses, lymphatic invasion was a significant independent predictor of BCR in the overall patient group and was independently associated with BCR, even in patients without lymph node metastasis. In conclusion, evaluation of lymphatic invasion could be useful in predicting BCR in patients undergoing RARP and eLND for high-risk PC.

## 1. Introduction

Prostate cancer is the second most common cause of tumors and the fifth leading cause of cancer mortality [1]. Radical prostatectomy is a standard treatment method for localized prostate cancer (PC) and has been shown to have cancer-specific survival benefits compared with watchful waiting [2]. Robot-assisted radical prostatectomy (RARP) has become a preferred treatment choice for localized PC as an alternative to open radical prostatectomy (ORP) or laparoscopic radical prostatectomy (LRP) [3]. Advantages of this treatment include shallow learning curves, low blood loss, low transfusion rate and short hospitalization duration [4,5]. Additionally, the oncological and functional outcomes of RARP are equivalent to those of ORP or LRP [4].

Despite the therapeutic efficacy of radical prostatectomy for localized PC, biochemical recurrence (BCR) has been reported in 10–25% of patients within five years after radical prostatectomy [6,7,8,9,10,11]. BCR could lead to the development of clinical metastases and cancer mortality with a median duration of 8 years from BCR to metastases and 5 years from metastases to death [12]. The National Comprehensive Cancer Network (NCCN) risk classification is a representative risk classification that is used clinically for decisions with respect to treatment policies [13]. In a recent study, five- and eight-year BCR-free rates in patients who underwent radical prostatectomy and pelvic lymph node dissection for NCCN high-risk PC were 47.8% and 39.6%, respectively [13].

Although attention should be paid to BCR after radical prostatectomy, especially in patients with high-risk PC, the risk of BCR may vary depending on histopathological findings of resected prostate specimens. Several risk factors, including extracapsular spread and positive surgical margin, are traditionally considered with respect to BCR after radical prostatectomy [14]. Moreover, lymph node metastasis has been reported to be a significant predictor of unfavorable postoperative progress, including BCR and cancer death [15,16,17]. Despite recent advances in imaging technologies, pelvic lymph node dissection remains the gold-standard technique for lymph node staging [18]. The direct therapeutic effect of lymph node dissection remains unclear, but it is generally accepted that extended lymph node dissection (eLND) provides appropriate prognostic information to assist with follow-up after surgery [19,20]. On the other hand, the presence of microlymphatic invasion in resected prostate specimens could be a useful predictor of BCR after surgery [14,21]. However, evidence of the prognostic impact of microlymphatic invasion is still limited, and, to the best of our knowledge, its predictive power for BCR in patients who have undergone radical prostatectomy and eLND for high-risk PC remains unclear. The aim of this prospective cohort study was to examine the association between microlymphatic invasion and postoperative BCR in patients who underwent RARP with eLND for NCCN high-risk PC.

## 2. Materials and Methods

This single-institutional prospective cohort study includes the 183 consecutive patients with NCCN high-risk PC who underwent RARP with eLND between June 2014 and August 2019. According to the NCCN guidelines, patients with high-risk PC are those with at least one of the following factors: clinical T stage 3a, maximum biopsy Gleason score of 8–10 and initial serum prostate serum antigen (PSA) >20 ng/mL [13]. The study was approved by the Wakayama Medical University Hospital Institutional Review Board (approval number 1670), and informed consent was preoperatively obtained from all subjects involved in the study.

We prospectively recorded preoperative patient backgrounds, including age, body mass index (BMI), smoking history, initial serum PSA, PSA density, maximum biopsy Gleason score, number of biopsy-positive cores and clinical stage. Initial serum PSA was defined as the PSA value just before prostate biopsy. PSA density was calculated by dividing the initial serum PSA by the prostate volume based on ultrasound examination. We also postoperatively recorded histopathological findings of resected specimens, including maximum Gleason score, pathological T stage, extraprostatic extension (EPE), seminal vesicle invasion (sv), lymphatic invasion (ly), venous invasion (v), resected margin (RM) and pathological N stage. Postoperative serum PSA value was evaluated and recorded every three months in the two years after surgery and every six months thereafter. BCR was defined as elevation of postoperative PSA > 0.20 ng/mL. If postoperative PSA was not reduced to <0.2 ng/mL, the operation date was taken as the recurrence date.

Three surgeons (I.H., Y.K. and K.K.), who are all experienced and skilled at ORP, LRP and RARP, performed RARP using a standard six-port, transperitoneal technique, employing the da Vinci Si system or da Vinci Xi system (Intuitive Surgical Inc., Sunnyvale, CA, USA) [22,23]. Our eLND template consisted of common iliac up to the ureteric crossing, external and internal iliac and obturator. eLND was performed, followed by cutting of the endopelvic fascia, transection of the bladder neck and dissection of the seminal vesicle and the prostate in an antegrade fashion. Nerve sparing was performed on the side, which met the following requirements according to the policies of our department: (1) non-detection of tumor by digital rectal examination, (2) no observation of disease lesion in the peripheral zone by magnetic resonance imaging and (3) two or fewer positive biopsy cores. After posterior musculofascial reconstruction and periurethral suspension stitching, vesicourethral anastomosis was performed with a running barbed suture. Resected specimens, including prostate and lymph nodes, were evaluated by two genitourinary pathologists, both of whom have more than 10 years of experience.

All statistical analyses were performed using JMP Pro 14, (SAS Institute Inc., Cary, NC, USA). Chi-square test and Mann-Whitney U test were used to compare patient demographics and histopathological findings of resected specimens between patients with ly0 and those with ly1. The Kaplan–Meier method and log-rank test were used to estimate the BCR rate and to compare the rates between groups. By using Cox proportional hazards regression analyses, univariable analyses and two models of multivariable analyses were performed to identify predictors of BCR. In model 1, Gleason-grade group of resected specimens, pathological T stage, RM, venous invasion (v) and pathological N stage were selected as predictors. In model 2, lymphatic invasion (ly) was used as a predictor in place of pathological N stage.

## 3. Results

### 3.1. Patient Demographics and Histopathological Findings of Resected Specimens

Preoperative patient demographics are summarized in Table 1. The median age was 69 years (quartile: 66–72 years). The median initial PSA was 10.9 ng/mL (quartile: 7.5–17.1 ng/mL). The biopsy Gleason-grade group was 4 in 110 patients (60%) and 5 in 42 patients (23%). Clinical T stage was T3 in 42 patients (23%).

Lymphatic invasion in the resected specimen was observed in 47 patients (26%). Table 2 shows a comparison of patient demographics and histopathological finding of resected specimens between patients with lymphatic invasion (ly1 group, *n* = 47) and without lymphatic invasion (ly0 group, *n* = 136). There was no significant difference in initial PSA, but the distribution of Gleason-grade group and pathological T stage was significantly different between the groups (*p* < 0.01 and *p* < 0.01, respectively), and the proportions of high Gleason grade and advanced pathological T stage in the ly1 group were higher than those in the ly0 group. Moreover, the percentages of EPE1, sv1,v1 and RM1 in the ly1 group were also higher than those in the ly0 group (*p* < 0.01, *p* < 0.01, *p* < 0.01 and *p* < 0.01, respectively). The median number of dissected lymph nodes was 21 (quartile: 14–28), and the percentage of nodal metastasis (pN1) in the ly1 group was higher than that in the ly0 group (26% vs. 4%, *p* < 0.01). Lymphatic invasion was observed in 35 patients (21%) in the pN0 group and 12 patients (71%) in the pN1 group (*p* < 0.01).

### 3.2. BCR Rates

The median postoperative follow-up period was 20 months (quartile: 10–34 months). During the follow-up period, BCR was observed in 48 patients (26%), and 1-year, 2-year and 3-year BCR rates were 18.0%, 27.5% and 32.6%, respectively.

A comparison of BCR rates according to the presence of nodal metastasis (pathological N0 vs. N1) and lymphatic invasion (ly0 vs. ly1) is shown in Figure 1. The BCR rate in the pN1 group was significantly higher than that in the pN0 group (*p* < 0.01) (Figure 1A). Moreover, the BCR rate in the ly1 group was significantly higher than that in the ly0 group (Figure 1B) (*p* < 0.01).

A comparison of BCR rates according to the combination of nodal metastasis and lymphatic invasion is shown in Figure 2. The BCR rates were significantly different among the groups (*p* < 0.01). The BCR rate in the pN1/ly0 and pN1/ly1 groups was higher than that in the pN0/ly0 group. The BCR rate in the pN0/ly1 group was also higher than that in the pN0/ly0 group.

### 3.3. Predictive Value of Lymphatic Invasion for BCR

The results of univariable analyses and two models of multivariable analyses of associations between predictive factors and BCR in overall patients are shown in Table 3. In model 1, significant independent predictors of BCR were venous invasion (v1), resected margin-positive (RM1) and nodal metastasis (pN1). In model 2, venous invasion (v1) and lymphatic invasion (ly1) were significant independent predictors of BCR.

The results of multivariable Cox proportional regression analyses of associations between predictive factors and BCR in only patients with pN0 are shown in Table 4. For patients without nodal metastasis, lymphatic invasion (ly1) was a significant independent predictor of BCR, as well as venous invasion (v1).

## 4. Discussion

In this prospective cohort study, we evaluated the association between microlymphatic invasion and postoperative BCR in patients who underwent RARP with eLND for NCCN high-risk PC. Lymphatic invasion was shown to be independently associated with BCR both in the overall patient group and in patients without lymph node metastasis. Although several previous studies have focused on the prognostic impact of microlymphatic invasion in patients who underwent radical prostatectomy, to the best of our knowledge, there has been no investigation into the impact in patients who underwent eLND and radical prostatectomy [14,21]. Moreover, the prognostic impact of microlymphatic invasion has not been properly evaluated in a prospective study. Therefore, this is the first study to prospectively examine the association between microlymphatic invasion and postoperative BCR in patients who underwent RARP with eLND.

Lymph node metastasis is known to be one of the most important prognostic factors and is associated with a high BCR rate and high risk of death from PC [15,16,17]. The direct therapeutic effect of lymph node dissection remains controversial. eLND may reduce the BCR rate and improve survival [11,16,24,25]. However, a recent systematic review showed that there was inadequate evidence of a direct therapeutic effect of eLND [19]. Moreover, in two recent randomized, controlled trials, no significant difference was observed in oncological outcomes, including BCR rate, between the limited lymph dissection group and the eLND group [20,26]. eLND could still have a therapeutic effect in certain patients with high-risk PC, and large-scale studies are expected [20,27,28]. On the other hand, another role of lymph node dissection is to provide exact pathological information about lymph node metastasis. The pathological diagnosis of lymph node metastasis offers appropriate prognostic information and assists with follow-up after surgery. eLND for high-risk PC could improve pathological staging and remove more metastatic lymph nodes compared to limited lymph node dissection [29]. Moreover, a recent randomized, controlled trial showed that eLND for intermediate and high-risk PC achieved improved pathological staging [20]. Performance of eLND for optimal staging in patients who require lymph node dissection has therefore been widely accepted. Although selection criteria for patients who require eLND have not yet been established, it is often practically performed in patients who are strongly suspected to have lymph node metastasis according to various nomograms or intermediate- and/or high-risk patients as classified by D’Amico risk stratification or NCCN risk stratification [30,31,32,33,34]. In the present study, we performed eLND, as well as RARP, in patients with NCCN high-risk PC.

In addition to traditional risk factors, including lymph node metastases, lymphovascular invasion (LVI) in prostate specimens has been widely reported to have a prognostic impact in patients who underwent radical prostatectomy. LVI is recommended as part of the standard examination of radical prostatectomy specimens by the International Society of Urological Pathology and defined as the presence of tumor cells in the endothelium-lined space [35]. In previous systematic reviews, LVI was suggested to be associated with a higher risk of BCR in patients who have undergone radical prostatectomy [35,36]. On the other hand, several recent studies evaluated lymphatic invasion separately from vascular invasion and investigated the prognostic impact of microlymphatic invasion. In a study that included 299 radical prostatectomy cases, Okubo et al. showed that lymphatic invasion was independently associated with lymph node metastasis, whereas venous invasion was not. They recommended that lymphatic and venous invasion should be evaluated separately rather than being combined into the category of LVI [37]. Wilczak et al. retrospectively analyzed pathological and clinical data from 14,528 consecutive patients who underwent radical prostatectomy and examined of the prognostic value of lymphatic invasion [21]. They concluded that evaluation of lymphatic invasion could provide comparable prognostic information to that of lymph node analysis. However, the study included a combination of patients who did and did not undergo lymph node dissection. Hashimoto et al. retrospectively reviewed 1096 patients with pT2 PC and RM negativity and investigated predictors for BCR after RARP [14]. Microlymphatic invasion was found to be an independent predictor of BCR, whereas microvascular invasion was not. However, their study also included patients who both did and did not undergo lymph node dissection.

In the present study, we prospectively evaluated the prognostic impact of microlymphatic invasion in patients who underwent RARP and eLND for high-risk PC. In our cohort, venous invasion and lymphatic invasion were observed in 11 and 47 patients, respectively. As shown in Table 3 and Table 4, whereas venous invasion was also related to BCR, lymphatic invasion was significantly associated with BCR independently of venous invasion. The association between lymph node metastasis and lymphatic invasion is summarized in Table 5. Lymph node metastasis (pN1) was observed in 17 patients in our cohort (9%) and was significantly associated with BCR after RARP. Moreover, lymph node metastasis was found to be an independent predictor of BCR. These results are consistent with those of previous studies. On the other hand, microlymphatic invasion (ly1) was observed in 12 patients with pN1 and 35 patients with pN0. Five patients with ly0 (5/136, 3.7%) had lymph node metastasis. Wilczan et al. reported that 4.3% of patients with ly0 had lymph node metastasis [21]. These discrepancies are considered to be due to microlymphatic invasion being overlooked. Complete detection of microlymphatic invasion is difficult, even with immunohistochemical staining by endothelial markers. eLND is therefore considered to have a diagnostic role in patients with high-risk PC. However, microlymphatic invasion was also found to be an independent predictor of BCR both in the overall patient group and in patients without lymph node metastasis. Thirty-five patients with pN0 (35/166, 21%) had microlymphatic invasion in our cohort. Although the possibility cannot be ruled out that lymph node metastasis was overlooked because of a technical problem associated with eLND or pathological evaluation in some of these patients, we consider microlymphatic invasion to be a premonitory finding of lymph node metastasis and a useful predictor of BCR in patients who underwent RARP and eLND for high-risk PC.

The current study is subject to several limitations. First, despite the advantage of being a prospective study, the sample was relatively small because the subjects were limited to patients who underwent RARP and eLND for high-risk PC. Second, the follow-up period after surgery was relatively short. Nonetheless, we believe that our study provides important insights into BCR after RARP and eLND for high-risk PC. To overcome these limitations and verify our results, a further large-scale prospective study with a long follow-up period is required.

## 5. Conclusions

Lymphatic invasion was shown to be a significant independent predictor of BCR after RARP and eLND in patients with high-risk PC. Even in patients without lymph node metastasis, lymphatic invasion was independently associated with BCR. Evaluation of lymphatic invasion could be useful for predicting BCR in patients who have undergone RARP and eLND for high-risk PC.

## Figures and Tables

**Figure 1 cancers-14-03466-f001:**
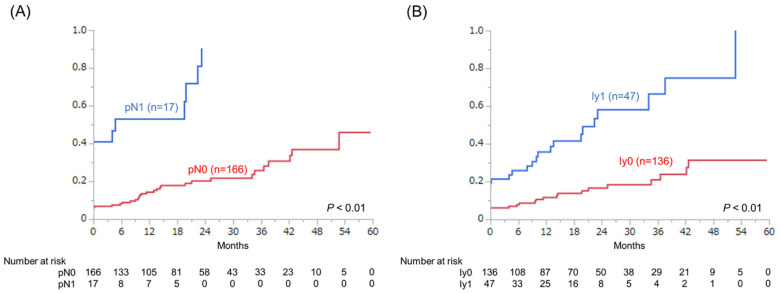
Kaplan–Meier analyses of BCR rates according to the presence of (**A**) nodal metastasis and (**B**) lymphatic invasion.

**Figure 2 cancers-14-03466-f002:**
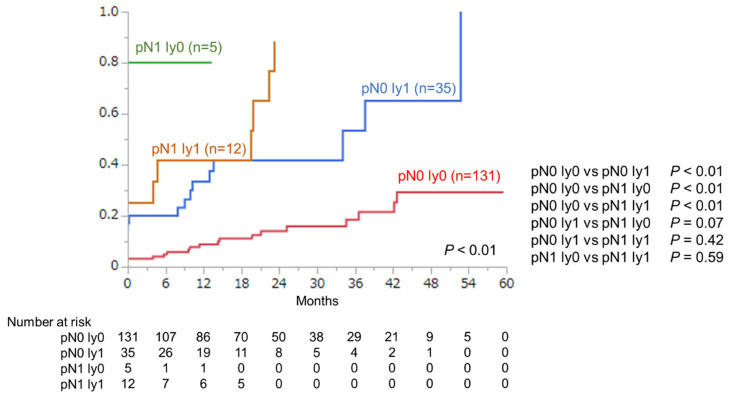
Kaplan–Meier analyses of BCR rates according to the combination of nodal metastasis and lymphatic invasion.

**Table 1 cancers-14-03466-t001:** Summary of preoperative patient demographics.

Age, years	69 (66–72) ^1^
BMI, kg/m^2^	24.1 (22.0–26.2) ^1^
Smoking history, *n* (%)	120 (66)
PSA, ng/mL	10.9 (7.5–17.1) ^1^
PSA density	0.43 (0.26–0.67)^1^
Biopsy Gleason-grade group, *n* (%)	
1	4 (2)
2	10 (6)
3	17 (9)
4	110 (60)
5	42 (23)
Clinical T stage, *n* (%)	
T1c	16 (9)
T2	125 (68)
T3	42 (23)

^1^ Continuous variables are shown in median (quartile) form.

**Table 2 cancers-14-03466-t002:** Comparison of patient demographics and histopathological findings of resected specimens between the ly0 and ly1 groups.

	ly0 Group (*n* = 136)	ly1 Group (*n* = 47)	*p* Value
Age, years ^1^	69 (65–72)	69 (66–72)	0.94
PSA, mg/dL ^1^	10.3 (7.0–17.0)	11.9 (8.6–17.6)	0.22
PSA density ^1^	0.42 (0.25–0.66)	0.47 (0.30–0.74)	0.28
Gleason-grade group, *n* (%)			<0.01
2	38 (28)	6 (13)	
3	59 (44)	17 (36)	
4	12 (9)	5 (11)	
5	26 (19)	19 (40)	
Pathological T stage, *n* (%)			<0.01
pT2	74 (54)	12 (26)	
pT3a	49 (36)	16 (34)	
pT3b	13 (10)	19 (40)	
EPE1, *n* (%)	58 (43)	31 (66)	<0.01
sv1, *n* (%)	13 (10)	19 (40)	<0.01
v1, *n* (%)	3 (2)	8 (17)	<0.01
RM1, *n* (%)	31 (23)	20 (43)	<0.01
pN1, *n* (%)	5 (4)	12 (26)	<0.01

^1^ Continuous variables are shown in median (quartile) form.

**Table 3 cancers-14-03466-t003:** Univariable and multivariable analyses of associations between various parameters and BCR in the overall patient group.

Univariable Analyses				Multivariable Analyses					
				Model 1			Model 2		
	HR	95% CI	*p* Value	HR	95% CI	*p* Value	HR	95% CI	*p* Value
Age, years	0.98	0.93–1.03	0.41						
PSA, ng/mL	1.02	0.98–1.05	0.18						
PSA density	1.22	0.57–2.36	0.57						
Gleason-grade groups 4–5 (vs. 2–3)	2.48	1.40–4.39	<0.01	1.70	0.90–3.18	0.09	1.81	0.97–3.38	0.06
pT3 (vs. pT2)	3.13	1.62–6.02	<0.01	1.95	0.96–3.96	0.06	1.94	0.96–3.89	0.06
v1 (vs. v0)	5.57	2.67–11.61	<0.01	2.73	1.19–6.24	0.01	2.58	1.13–5.84	0.02
RM1 (vs. RM0)	2.17	1.22–3.86	<0.01	1.92	1.06–3.48	0.03	1.70	0.93–3.10	0.08
pN1 (vs. pN0)	6.34	3.24–12.36	<0.01	2.73	1.29–5.75	<0.01			
ly1 (vs. ly0)	4.11	2.67–11.61	<0.01				2.33	1.22–4.42	0.01

**Table 4 cancers-14-03466-t004:** Multivariable analyses of associations between various parameters and BCR only in patients with pN0.

	HR	95% CI	*p* Value
Gleason-grade groups 4–5 (vs. 2–3)	2.03	0.99–4.11	0.05
pT3 (vs. pT2)	1.75	0.82–3.67	0.14
RM1 (vs. RM0)	1.93	0.95–3.91	0.06
v1 (vs. v0)	4.10	1.50–11.19	<0.01
ly1 (vs. ly0)	2.59	1.25–5.32	0.01

**Table 5 cancers-14-03466-t005:** Association between lymph node metastasis (pN) and microlymphatic invasion (ly).

		Lymph Node Metastasis, *n*		
		pN0	pN1	Total
Micro-lymphatic invasion, *n*	ly0	131	5	136
	ly1	35	12	47
	Total	166	17	183

## Data Availability

Not applicable.

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
