# Peer review of "Prognostic Impact of Lymphatic Invasion in Patients with High-Risk Prostate Cancer after Robot-Assisted Radical Prostatectomy and Extended Lymph Node Dissection: A Single-Institution Prospective Cohort Study"

_cancers, 2022, doi:10.3390/cancers14143466_

Round 1
Reviewer 1 Report
The manuscript by Yamashita et.al. explored the impact of microscopic lymphatic invasion on biochemical recurrent in a cohort of patients with NCCN high risk prostate cancer who was treated with radical prostatectomy and extended lymph nodes dissection. While this is a topic of potential interest to the urological oncology community, there are several concerns regarding statistics used in the manuscript.
1. The possibility of micro-lymphatic invasion as a confounder of nodal metastasis on BCR needs to be addressed. This concern is raised from data presented in Table 5 - a chi square test showed a p < 0.05 suggesting association between these two clinical variables. As such, the reported predictable effect of MLY might be just from the fact patients with nodal metastasis are more likely to experience BCR, which has been known for a long time in the field.
2. Table 3 presented results regression results on the association of BCR with multiple variables. For regression studies the rule of thumb is 10 events are needed for each regression variable to avoid over fitting of data (eg. PMID 8970487). For each model the authors included 9 variables (categorical variable with k levels requires k-1 indicator variables) so the minimal number of events required is 90. However only 48 patients in the cohort experienced BCR. The authors are encouraged to either reduce the number of regression variables or provide justification of sample size choice based on power analysis.
3. Figure 2- besides overall p value, please provide results from significance test of each pair-wire comparisons with appropriate p value corrections.
Minor comment
1. Please provide description of LN dissection template (i.e. dissection boundaries) of the eLND used in the study.
Author Response
The manuscript by Yamashita et.al. explored the impact of microscopic lymphatic invasion on biochemical recurrent in a cohort of patients with NCCN high risk prostate cancer who was treated with radical prostatectomy and extended lymph nodes dissection. While this is a topic of potential interest to the urological oncology community, there are several concerns regarding statistics used in the manuscript.
- The possibility of micro-lymphatic invasion as a confounder of nodal metastasis on BCR needs to be addressed. This concern is raised from data presented in Table 5 - a chi square test showed a p < 0.05 suggesting association between these two clinical variables. As such, the reported predictable effect of MLY might be just from the fact patients with nodal metastasis are more likely to experience BCR, which has been known for a long time in the field.
We appreciate your important comments.
As you pointed out, lymph nodal metastasis has been traditionally known as a significant predictor of BCR. In addition, micro-lymphatic invasion is a strong confounder of lymph nodal metastasis, as shown in Table 5. However, while only 17 patients had lymph nodal metastasis, lymphatic invasion was observed in 47 patients. In addition, as shown in Table 4, lymphatic invasion was an independent predictor of BCR in patients without lymph nodal metastasis.
These results suggest that the predictable effect of lymphatic invasion does not simply reflect the fact that patients with nodal metastasis are more likely to experience BCR.
- Table 3 presented results regression results on the association of BCR with multiple variables. For regression studies the rule of thumb is 10 events are needed for each regression variable to avoid over fitting of data (eg. PMID 8970487). For each model the authors included 9 variables (categorical variable with k levels requires k-1 indicator variables) so the minimal number of events required is 90. However only 48 patients in the cohort experienced BCR. The authors are encouraged to either reduce the number of regression variables or provide justification of sample size choice based on power analysis.
Thank you for your important suggestions.
According to your suggestions, we have reduced the number of variables to five. We have also added univariable analyses. Sentences in the Materials and Methods section were therefore revised as follows:
“By using Cox proportional hazards regression analyses, univariable analyses and two models of multivariable analyses were performed to identify predictors of BCR. In model 1, Gleason grade group of resected specimens, pathological T stage, RM, venous invasion (v) and pathological N stage were selected as predictors.” (page 3, lines 113-117)
Accordingly, Table 3 and Table 4 were revised, as shown in the revised manuscript. The explanation of the results of Table 3 and Table 4 have also been revised as follows:
“The results of univariable analyses and two models of multivariable analyses of associations between predictive factors and BCR in overall patients are shown in Table 3. In model 1, significant independent predictors of BCR were venous invasion (v1), resected margin positive (RM1) and nodal metastasis (pN1). On the other hand, iIn model 2, venous invasion (v1) and lymphatic invasion (ly1) were significant independent predictors of BCR.” (page 6, lines 165-169)
“For patients without nodal metastasis, lymphatic invasion (ly1) was an independent significant predictor of BCR as well as venous invasion (v1).” (page 6, lines 175-176)
- Figure 2- besides overall p value, please provide results from significance test of each pair-wire comparisons with appropriate p value corrections.
We appreciate your useful suggestion.
We have now provided the results from significance test or each pair-wise comparisons with P value corrections besides overall P value.
Minor comment
- Please provide description of LN dissection template (i.e. dissection boundaries) of the eLND used in the study.
Thank you for your important suggestion.
We have added the following sentence to the Materials and Methods section:
“Our eLND template consisted of common iliac up to the ureteric crossing, external and internal iliac and obturator.” (page 3, lines 98-99)
Reviewer 2 Report
The authors reported an association between biochemical recurrence and lymphatic invasion in patients who received RARP and eLND. Findings regarding lymphatic invasion have been previously reported. Because of the small size and short observation period of this cohort, it is necessary to consider reporting and analysis methods that highlight the unique features of this study.
Details of eLND (method, extent, and results of the number of lymph nodes removed) are not described. If this study is unique in that it is a cohort in which all patients underwent eLND, then details should be provided.
If pN and Lymphatic invasion are related, then the results of multivariate analysis in the same model that accounts for their confounding should be presented. It is important to know if lymphatic invasion has strength beyond lymph node metastasis, which has not been properly analyzed (Table 3).
The significance of analyzing venous and lymphatic invasion separately in eLND cases has not been demonstrated in this paper or previous studies, and if you are going to discuss it, please provide the results of your cohort.
Due to the small size and short observation period of this cohort, the Kaplan-Meier curve should be included "risk at time" or "censored".
Minor
BCR-free rate (Line 53)
Author Response
The authors reported an association between biochemical recurrence and lymphatic invasion in patients who received RARP and eLND. Findings regarding lymphatic invasion have been previously reported. Because of the small size and short observation period of this cohort, it is necessary to consider reporting and analysis methods that highlight the unique features of this study.
Details of eLND (method, extent, and results of the number of lymph nodes removed) are not described. If this study is unique in that it is a cohort in which all patients underwent eLND, then details should be provided.
We appreciate your important suggestion.
We have added the following sentence to the Materials and Methods section:
“Our eLND template consisted of common iliac up to the ureteric crossing, external and internal iliac and obturator.” (page 3, lines 98-99)
We have already described about the number of lymph nodes removed as follows:
Median number of dissected lymph nodes was 21 (quartile: 14-28), and the percentage of nodal metastasis (pN1) in ly1 group was higher than that in ly0 group (26% vs 4%, P < 0.01). (page 4, lines 136-139)
If pN and Lymphatic invasion are related, then the results of multivariate analysis in the same model that accounts for their confounding should be presented. It is important to know if lymphatic invasion has strength beyond lymph node metastasis, which has not been properly analyzed (Table 3).
Thank you for your valuable opinions. As shown in Table 5, lymphatic invasion was a strong confounder of lymph nodal metastasis. Although we tried to perform multivariable analysis by using both lymph nodal metastasis and lymphatic invasion in the same model, our statistical software showed that the model was unreliable. In addition, another reviewer suggested that we should have reduced the number of predictors included in multivariable analyses. Two models of multivariable analyses were therefore performed. In model 1, pathological N stage was used as a predictor. In model 2, lymphatic invasion was used as a predictor in place of pathological N stage. We also examined the impact of lymphatic invasion on BCR in patients without lymph nodal metastases.
Our results did not show that lymphatic invasion had stronger impact on BCR than lymph nodal metastasis, but showed that lymphatic invasion was significantly associated with BCR both in overall patients and patients without lymph nodal metastasis.
The significance of analyzing venous and lymphatic invasion separately in eLND cases has not been demonstrated in this paper or previous studies, and if you are going to discuss it, please provide the results of your cohort.
Thank you for your valuable suggestion.
The presence of venous invasion was also prospectively recorded. Data on venous invasion has therefore been added to Table 2. In addition, venous invasion was added to Cox proportional hazards regression analyses, as shown in revised Table 3 and Table 4. Revised sentences in the Materials and Methods section are as follows:
“By using Cox proportional hazards regression analyses, univariable analyses and two models of multivariable analyses were performed to identify predictors of BCR. In model 1, Gleason grade group of resected specimens, pathological T stage, RM, venous invasion (v) and pathological N stage were selected as predictors.” (page 3, lines 113-117)
Description of the results of Table 3 and Table 4 was revised as follows.
“The results of univariable analyses and two models of multivariable analyses of associations between predictive factors and BCR in overall patients are shown in Table 3. In model 1, significant independent predictors of BCR were venous invasion (v1), resected margin positive (RM1) and nodal metastasis (pN1). In model 2, venous invasion (v1) and lymphatic invasion (ly1) were significant independent predictors of BCR.” (page 6, lines 165-169)
“The results of multivariable Cox proportional regression analyses of associations between predictive factors and BCR in only patients with pN0 are shown in Table 4. For patients without nodal metastasis, lymphatic invasion (ly1) was an independent significant predictor of BCR as well as venous invasion (v1).” (page 6, lines 173-176)
In our cohort, venous invasion and lymphatic invasion were observed in 11 and 47 patients, respectively. While venous invasion was also associated with BCR, lymphatic invasion was associated with BCR independently from venous invasion. We therefore added the following sentence to the Discussion section:
“In our cohort, venous invasion and lymphatic invasion were observed in 11 and 47 pa-tients, respectively. As shown in Table 3 and Table 4, while venous invasion was also related to BCR, lymphatic invasion was significantly associated with BCR inde-pendently from venous invasion.” (page 8, line 239-page 8, line 242)
Due to the small size and short observation period of this cohort, the Kaplan-Meier curve should be included "risk at time" or "censored".
We appreciate your valuable suggestion.
We added the number at risk to Figure 1 and Figure 2.
Minor
BCR-free rate (Line 53)
Thank you for pointing out our mistake.
We revised “BCR rates” to “BCR-free rates”. (page 2, line 53)
Reviewer 3 Report
Aim of this manuscript is to assess the prognostic impact of lymphatic invasion in patients with high-risk localized prostate cancer treated with RARP and pelvic lymphnode dissesction. Aim of the study is clear and relevant; however, there are some inherent flaws that strongly affect the reliability of the results.
-Firstly, despite the prospective design of the study, the low numbers of patients included (183 in a five-year period) does not allow to draw strong and reliable conclusions. Moreover, several pre-operative tumor-related and patient-related features are missing including for example PSA density, smoking status of patients etc.
-No information regarding numbers of pathologists involved and their experience is available. However, in this study pathological examination is paramount.
-MRI is reported in Table and in Multivariable analyses; however, I did not understand the meaning of its inclusion as RM1 (vs RM0). Furthermore, it would have been of interest to investigate predictors of lymphatic invasion at MRI imaging.
-Overall, the manuscript lacks of originality and there are already several studies reporting similar results.
Author Response
Aim of this manuscript is to assess the prognostic impact of lymphatic invasion in patients with high-risk localized prostate cancer treated with RARP and pelvic lymphnode dissesction. Aim of the study is clear and relevant; however, there are some inherent flaws that strongly affect the reliability of the results.
-Firstly, despite the prospective design of the study, the low numbers of patients included (183 in a five-year period) does not allow to draw strong and reliable conclusions. Moreover, several pre-operative tumor-related and patient-related features are missing including for example PSA density, smoking status of patients etc.
Thank you for your important comments.
As described in the limitations section, although it had the advantage of being a prospective study, the sample number was relatively small because this was single institutional study and the subjects were limited to patients who underwent RARP and eLND for their high-risk PC. To verify our results, further large-scale prospective study with a long follow-up period is therefore planned.
PSA density and smoking history were also prospectively recorded. We have revised the sentence in the Materials and Methods section as follows:
“We prospectively recorded preoperative patient backgrounds including age, body mass index (BMI), smoking history, initial serum PSA, PSA density, maximum biopsy Gleason score, number of biopsy positive cores, and clinical stage.” (page 2, lines 83-85)
We added the following sentence to the Materials and Methods section:
“PSA density was calculated by dividing initial serum PSA by prostate volume based on ultrasound examination.” (page 2, lines 86-87)
In addition, the data about smoking history was added to Table 1 and the data about PSA density was added to Table 1, Table 2, Table 3 and Table 4.
-No information regarding numbers of pathologists involved and their experience is available. However, in this study pathological examination is paramount.
Thank you for your valuable suggestions. We have added the following sentence to the Materials and Methods section:
“Resected specimens including prostate and lymph nodes were evaluated by two geni-tourinary pathologists who each have more than 10 years of experience." (page 3, lines 107-108)
-MRI is reported in Table and in Multivariable analyses; however, I did not understand the meaning of its inclusion as RM1 (vs RM0). Furthermore, it would have been of interest to investigate predictors of lymphatic invasion at MRI imaging.
Thank you for your important comments. Positive surgical margin in resected prostate specimen has been traditionally reported to be an important risk factor of BCR, so we added it to our analyses as RM1 (vs RM0).
Although we analyzed the association between pathological findings of resected specimen and BCR in the present study, various previous studies have examined whether the impact of various preoperative MRI parameters on BCR. In addition, as you pointed out, it is very interesting to investigate the association between preoperative MRI parameters and lymphatic invasion in resected specimen. Future study will therefore focus on preoperative MRI parameters.
-Overall, the manuscript lacks of originality and there are already several studies reporting similar results.
We appreciate your valuable opinion. While several previous studies have already investigated the prognostic impact of lymphatic invasion, no studies have yet focused on the association between lymphatic invasion and biochemical recurrence rates in patients who underwent eLND as well as radical prostatectomy for their high-risk prostate cancer. In addition, this is a prospective cohort study. For these reasons, we believe that our study adds important information to the literature.
Round 2
Reviewer 1 Report
Concerns are addressed by authors and the quality of presentation is improved.
Reviewer 2 Report
The authors have revised the manuscript more clearly and addressed concerns raised by reviewers. The manuscript would be informative for readers.
Reviewer 3 Report
The issues raised during the review process have been addressed by Authors; the overall quality of the manuscript has been improved.